# A Cost-Effective, Integrated Haptic Device for an Exoskeletal System

**DOI:** 10.3390/s22239508

**Published:** 2022-12-05

**Authors:** Maciej Rećko, Kazimierz Dzierżek, Rafał Grądzki, Jozef Živčák

**Affiliations:** 1Faculty of Mechanical Engineering, Bialystok University of Technology, Wiejska st. 45C, 15-351 Białystok, Poland; 2Faculty of Mechanical Engineering, Technical University of Kosice, Letná st. 1/9, 042-00 Košice, Slovakia

**Keywords:** integrated haptic device, electromagnet, force feedback, passive haptic system

## Abstract

The paper presents an innovative integrated sensor-effector designed for use in exoskeletal haptic devices. The research efforts aimed to achieve high cost-effectiveness for a design assuring proper monitoring of joint rotations and providing passive force feedback. A review of market products revealed that there is space for new designs of haptic devices with such features. To determine the feasibility of the proposed solution, a series of simulations and experiments were conducted to verify the adopted design concept. The focus was set on an investigation of the force of attraction between one and two magnets interacting with a steel plate. Further, a physical model of an integrated joint was fabricated, and its performance was evaluated and compared to a similar commercially available device. The proposed solution is cost-effective due to the use of standard parts and inexpensive components. However, it is light and assures a 19 Nm braking torque adequate for the intended use as a haptic device for upper limbs.

## 1. Introduction

Human cognitive functions are necessary for applications requiring advanced operations and missions of remotely controlled robots. Among the advantages of direct control over a robot’s operation are high precision and the possibility of teleoperation using a vision system that a robot can be equipped with. Robots with such systems are widely used in search and rescue missions, disaster relief missions, distant areas exploration and discovery missions, and research missions. These tasks require a device that enables remote control over the robot’s actions.

Exoskeleton devices are used for remote control [1,2,3,4]. The task is to collect information about the positions of the operator’s hand and convert this data into signals controlling the movements of manipulators. It is possible to use several haptic solutions to increase the number of stimuli affecting the operator’s control scheme over robots. For example, an operator can feel the weight of the objects he manipulates, surface texture, and, in some cases, temperature. All these data increases the immersion, allowing the operator to have a much better sense of the state of the environment and the object in the workspace.

Literature indicates three main areas of application of haptic devices [5,6,7]: (1) displays that map, for example, the shape of the surface [8,9], (2) parallel systems that allow the representation of forces exerted, for example, on the tool [10,11], and (3) wearable tools acting on the operator [12,13,14]. They are designed with various physical phenomena and principles of operation.

Magnetorheological fluid subjected to a magnetic field changes its physical properties. Mainly, an increase in its apparent viscosity occurs [15]. It is widely used to precisely control the damping parameters of smart dampers [16]. This fluid can also be used for brakes and haptic devices [17,18]. The ability to control braking torque using only electrical signals helps create haptic feedback. One can also control fluid flow, thus obtaining damping and braking action accordingly. Using an electro-hydraulic effector [19] takes advantage of extremely low compression tendency in liquids, providing an accurate rendering of force feedback stimuli. Pneumatic systems are also found among haptic drive solutions. They use case covers for specific devices, such as wrist effectors [13,19] or haptic buttons [20,21].

An interesting approach is based on a pulley system used to achieve haptic feedback [22]. Oscillators offer a different approach to haptic feedback. Slight mass movements in rapid succession can reliably inform users about various states of controlled machinery. The most commonly used oscillators are linear [23,24] and rotary [25]. Such oscillators are used in commercial solutions [26,27].

The back driveability of a motor can also be used to achieve force feedback. The approach differs based on the type of motor used and the gearbox or controller. Force is provided to the shaft, and the motor counteracts such a force directly and adequately. Examples of haptic feedback context use of this method are given in [28,29,30,31]. 

A magnetic field can also be a source of force feedback. For example, a solenoid can govern the movement of a core traversing along the axis of a device, thus exerting force [32]. This scenario can also be reversed, with the solenoid being a moving part [33]. Both approaches swiftly achieve the desired movement along the actuator’s axis. The undisputed advantage of this approach is the ability to use its reverse movement, externally forced, to monitor the value of the force applied. Unfortunately, the solenoid-based approach allows only linear movement, which is constrained by the device’s design.

For controlling rotational movements, eddy currents are frequently employed. Eddy currents allow damping and braking revolutions by applying a magnetic field to both sides of a rotating disk. The resulting electrodynamic force slows down or even halts the disk’s movement [34]. A more advanced approach, using Coulomb friction, was explored for scenarios requiring low-velocity applications [35]. It is also possible to provide damping force in even 3-DOFs accurately imitating ball joints [36]. Such solutions are found in haptic feedback devices [37,38,39]. They are integrated into HMI devices built to ensure the exchange of information between the user and the machine [40].

An attempt to solve the problem of simultaneously exerting force on the operator and monitoring the movement was made using intelligent servos. These devices are constructed using absolute encoders with a gearbox and a high-class electric drive. The torque offered by the exemplary device—the Dynamixel MX-64 servo [41]—is 6 Nm with a nominal power supply, i.e., 12 V. However, this device, due to the high ratio (200:1) and due to software limitations, cannot be subjected to such a back driving. 

Generally, attention should be paid to the potential limitations of the force applied to the operator’s limb from an external source. For example, applying it too abruptly or at too high a force can lead to discomfort and, in unfortunate cases, even cause injuries. Literature sources suggest two solutions to this problem. The first is to provide so-called passive force feedback. In such a scenario of operation, we only limit the operator’s movement by inhibiting the rotation of individual joints. This approach to the safety problem is suggested, especially in cases where movement is characterized by high dynamics and variability of these dynamics [42]. The second possible solution is constant monitoring of the operator’s condition using non-invasive methods. For example, blood flow parameters in the limb can be monitored [43]. Based on the data from the pulse oximeter, it is possible to analyze the rate of blood flow through blood vessels and the degree of its oxygenation. A disturbance of any of these parameters may indicate that too strong forces are exerted on the limb.

## 2. Proposed Concept, Constraints, and Assumptions

The objective of the present paper is an integrated joint with haptic feedback. The presented literature analysis allows for adopting design assumptions regarding such an integrated joint. In the development of its original construction, the assumption was made that haptic feedback would be achieved by passive means, i.e., inhibiting the operator’s movements.

The research leads toward designing an innovative, integrated haptic joint system capable of sensing joint rotation and providing force feedback. An additional goal was to create a device that was as cost-effective as possible. We aimed to achieve that by using off-the-shelf components and materials available.

In the design phase, we estimated that the sufficient braking torque of our solution should be 15 Nm. This value was based on analyses of the technical specifications of small-sized robotic arms with comparable torque joints. 

The project’s constraints derive from the proposed system’s wearable nature. For example, an integrated joint system cannot be too heavy or disproportionately large compared to a human limb. Therefore, the maximums acceptable for this project were set. Weight must not exceed 800 g. The device should fit within a cylinder of 200 mm in diameter and a height of 100 mm.

Contributing to our goal of providing passive force feedback, it was decided to focus on providing braking action with our device. A purely mechanical approach would require providing a source of movement in the form of a motor or a servo. To provide adequate force for braking action, the required motor would be relatively big and heavy. Based on the literature review, it was decided to use an electromagnet-based design due to its ease of implementation and control and acceptable power consumption. We have not found sources that would indicate that this approach was explored before. The challenge was to achieve high braking torque using this actor and combine its operation with rotation monitoring. To this end, it was decided to use an encoder, which was the best device for this task. We decided on a capacitive absolute encoder due to its indifference to operation in the proximity of a strong electromagnetic field. An additional requirement was that the joint should allow for uninterrupted rotation when not energized, allowing for the smooth operation of the device.

## 3. Commercially Available Solutions

A thorough examination of commercially available solutions allowing halting rotational movements showed several designs dedicated to similar applications. Most of them are either electromagnetic clutches or electromagnetic brakes that are designed for industrial applications. Their design favors the high rotational speed of operation and very high torque. Due to their intended use case, they are encased in large and heavy steel cases that provide rigidity and dissipate heat. The devices examined have features preventing their use in our case. For example, most devices have too large dimensions or are too heavy to be wearable. Their design does not meet our mass and dimensions restrictions. We found only two manufacturers offering relatively small and lightweight devices. However, they are very similar in design, so one of them was used as a reference design to compare with our results.

### Design of the Joint

Before designing the details of the joint, we performed a simulation of its actions using COMSOL software [44] for solving Maxwell equations [45], Faraday [46,47], and Ampere laws [48,49].

It was decided to simulate the operation of a singular electromagnet set on a ferromagnetic steel plate. The electromagnet parameters were obtained through measurements and catalog data for pre-selected items (Table 1).

The electromagnet connected to the 12 V power supply has a stable current of 0.6 A during operation. Additional parameters used in the simulation are shown in Table 2. Mesh used in modeling is characterized in Table 3 and depicted in Figure 1.

The simulation results are shown in Figure 2 and Figure 3 in the form of magnetic flux density distribution within the device. The magnetic flux is the strongest on the perimeter of the examined part, and it is found that it permeates approximately 5 mm distance from the steel plate surface.

Simulation results of attraction between the steel plate and one electromagnet are shown in Figure 4. The electromagnets’ position was assumed to be 0.1 mm from the steel plate. This value allows us to accommodate imperfections in the manufacturing process of both elements.

The mesh parameters for two magnets configurations are collected in Table 4.

The mesh density, depicted in Figure 5, was increased closer to the contact surface between electromagnets.

The magnetic flux density distributions for two magnets are depicted in Figure 6 and Figure 7, and attraction force as a function of distance from the plate is plotted in Figure 8.

The simulation results show that the force increases by roughly 10% for two magnets configuration. Such an increase clearly shows that the design criteria will not be reached by increasing the number of magnets. Since experimental measurements of the attraction force subsequently validated these findings, an alternative design concept was adopted, as described in the following chapters.

## 4. Design Based on the Increased Friction

Since increasing normal force was impractical, we focused on increasing the friction coefficient. According to literature data [50,51,52,53,54,55], the friction coefficient of a nickel-coated electromagnet on hard steel varies from 0.178 when lubricated to 0.6 in the case of dry friction. Since we cannot assure perfectly dry friction, we had to assume the worst-case scenario and use 0.178 as our base coefficient. In order to achieve the required torque of 15 Nm, we analyzed different materials to use as braking pads.

### 4.1. Modeling One Magnet Joint with Brake Pads for Increased Friction Force

The following assumptions were made for one electromagnet and friction pads design: (a) force of a single magnet on a steel plate is 728 N, (b) commercially available brake pads to be used (e.g., BARADINE 450 Brake shoes). The design of an integrated joint utilizing brake pads is depicted in Figure 9 and Figure 10. A disk pad and steel disc are fastened to the housing using bolts. Based on the simulation, the height of the steel disc is set to 5 mm. All elements except steel plate and ball bearings are made of aluminum alloy to dissipate heat and lower overall design mass.

### 4.2. The Principle of Operation

All elements of the joint are depicted in Figure 10. The electromagnet (3) pulls the steel plate (2) towards itself when energized. The steel plate is rigidly attached to the brake disk (5) using screws encased in sleeves. This setup is held together by ball-bearing housing (4) with bores to feed sleeved screws through and allow this component to move along the electromagnet’s axis. This movement pushes the brake disk (5) towards brake pads (6) fixed to the electromagnet’s housing (7). A ball bearing is firmly attached to an electromagnet’s brim and provides rotational movement ease. The encoder (1) is mounted on top of the steel plate, and a shaft is directly attached to the center of the electromagnet. To compensate for the up and down movement of the steel plate with brake disk component, a slight slack was accounted for movement along the measurement axis. This slack is incorporated into the design of an AMT 23 encoder, and our available motion range does not use all of the available 1 mm of slack.

## 5. Experimental Validation

In order to verify the assumptions made, a test system was built that allows examining the performance of different types of brake pads. The MTS Insight testing machine and the experimental rig are shown in Figure 11. The machine provided the force to rotate the device while measuring the displacement of the beam. The rotation of the device was monitored with an embedded encoder. This data correlated with displacement and provided information for braking torque plots. In addition, we were able to test the performance of an encoder while braking takes place.

The braking torque measured using the experimental setup is plotted in Figure 12 for three different materials as a function of the rotation angle. The best results were obtained using elastic rubber brake pads.

Further tests were performed for four braking speeds: 0.5 mm/s, 1 mm/s, 2.5 mm/s, and 5 mm/s. All tests were performed using the nominal voltage of an electromagnet—12 V with different current values in increments of 0.1 A. The results of the tests are depicted in Figure 13. The results clearly show that our solution provided a smooth braking curve along the whole test range in contrast to a commercial one that has tended to have bursts of torque alternating with skidding phases. 

Our test indicated that at the same current level, our solution provides twice as much braking torque in the operation point at a comparable speed. In addition, the torque increases with the speed of braking. This phenomenon is beneficial for the proposed use case as more dynamic movements might require more present force feedback that our device provides.

## 6. Summary

The designed joint presented here assures 19 Nm of braking torque. This value is sufficient for torques to be exerted by a limb’s action. Furthermore, respective contracting stimuli would be noticeable and provide valuable information for the operator through force feedback [56,57,58].

Compared to the research of like solutions from the literature, our solution provides several advantages crucial for its use in a wearable haptic device. First, our device is small and relatively lightweight compared to eddy-current-based solutions. Our final prototype is 100 mm in diameter and 70 in height, weighing 765 g.

Gosline and Hayward [37] reported that their device has 50 mm of effective disc radius (that would give as 100 mm in diameter), and this dimension does not include a toroidal electromagnet attached to the disk. The mass of the device was not mentioned, unfortunately. However, the authors mention using two re25 Maxon motors weighing 130 g each. We can assume that mass of both devices might be comparable. It is also difficult to directly compare braking torque due to using an integral of damping in their work. However, researchers achieved 4 mNms of damping, which is orders of magnitude away from our reported 19 Nm of torque.

A multipole solution based on Coulomb friction by Iqbal and Yi [35] exhibited braking torque of 1 Nm at a current of 2.5 A. On the other hand, we achieved 19 Nm at only 0.51 A. Their second work [36] is extremally interesting due to enabling damping movement on three axes with high torque of around 96.1 Nm about X-Y and 98 Nm around the Z-axis. However, those values were reached using 4 A of current. Since their report currents were indicated in increments of one, we can compare the non-energized state of 0 A (idle state) where reported torques are: about 4.3 Nm for X-Y and 4.9 Nm around Z-axis and for current of 1 A it was 15.9 Nm and 23 Nm respectively. Our solution is comparable to those solutions, although while idle (not energizged), our device provides free movement with no dumping.

It is also worth noting that our device is fully integrated with bearings, mounting points, and an encoder, ready to use in a haptic device without any modifications. Solutions of other researchers would require further work to implement them into complete haptic systems.

In addition, our design fits within the assumed constraints of mass and dimension. As a result, this device is slightly smaller than the commercial solution (diameter: 120 mm, height: 100 mm, mass: 770 g) while exhibiting higher braking performance.

A 19 Nm of braking torque value is comparable to the values achievable by commercial electromagnetic braking devices tested. It is also worth noting that the presented design is by far more cost-effective. Based on the market prices at the time of writing this paper, raw materials and components oscillate around USD 30 without an encoder. The price of a commercial product varies between USD 150 and 250 and still would require an implementation of an external encoder for integrating sensing and acting, fully integrated into the current design. We also observed the nominal operation of an encoder without any problems caused by braking action.

An additional advantage of the presented solution is its low power consumption. The power consumption of an integrated haptic joint is 12 W. Thermal energy generated by prolonged use of the electromagnet is well dispersed in the air because of using an all-aluminum housing.

The current communication describes a research effort that is a part of a more ambitious plan to build an exoskeletal phantom device to control robotic arms. We do not explore the possibility of application of the system for rehabilitation needs as we do not see an only passive device suitable for this use case. It is the first in the series of project progress reports. The design of a complete phantom device is depicted in Figure 14.

It consists of several links connected using integrated joints described in this paper. The kinematics of the solution does not directly follow the kinematic chain of a human arm. However, the presented device has the same range of movement and allows unrestricted operator motions. Furthermore, to give a broader range of information from the system, we use vibration motors on each link to inform the operator about approaching areas that are out of reach.

## Figures and Tables

**Figure 1 sensors-22-09508-f001:**
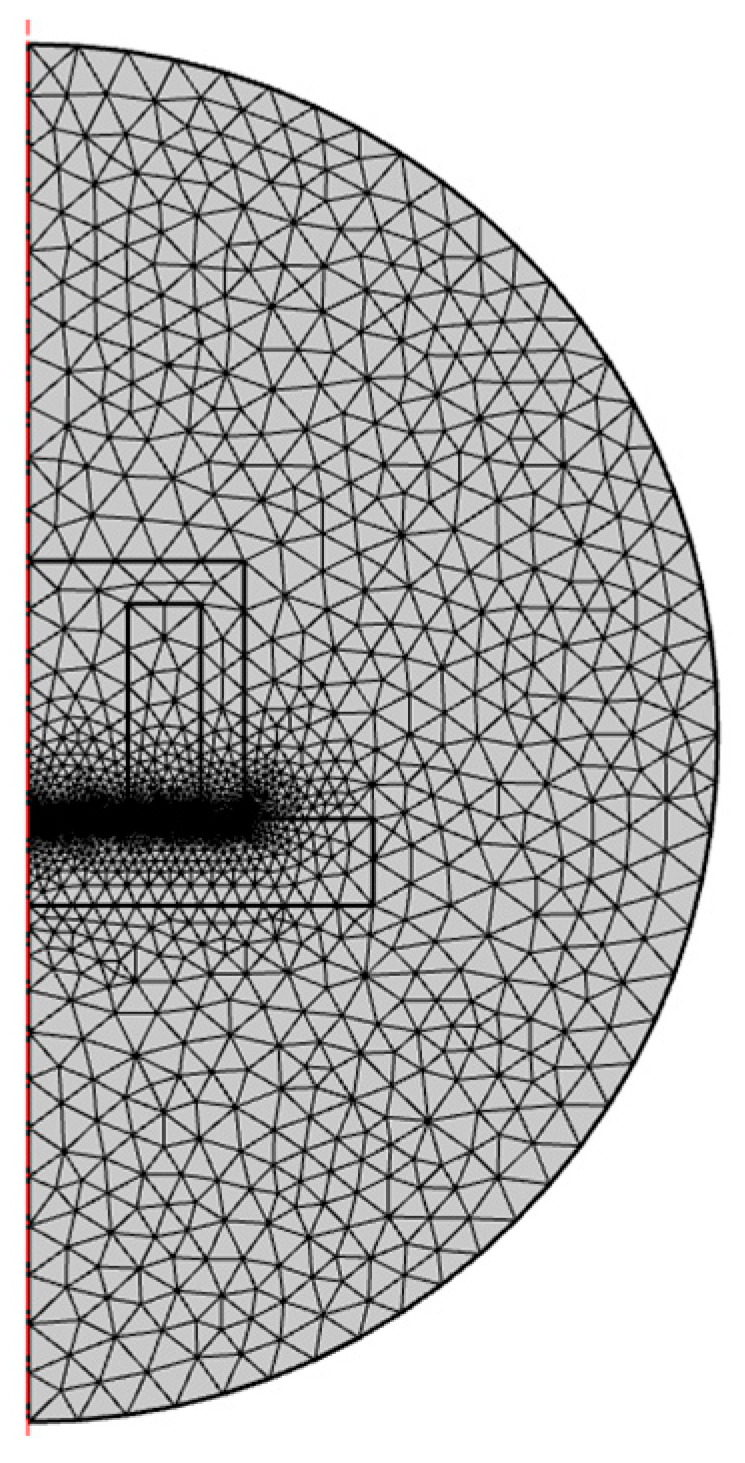
Measurement mesh.

**Figure 2 sensors-22-09508-f002:**
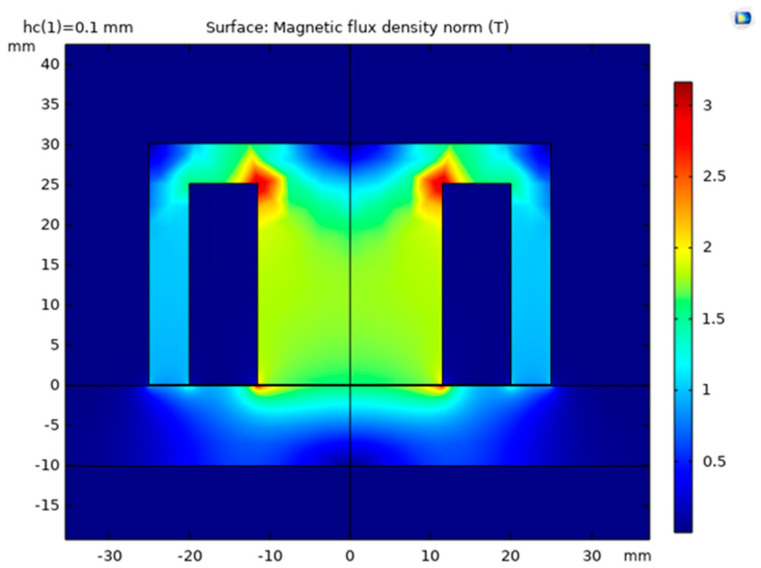
Surface magnetic flux density.

**Figure 3 sensors-22-09508-f003:**
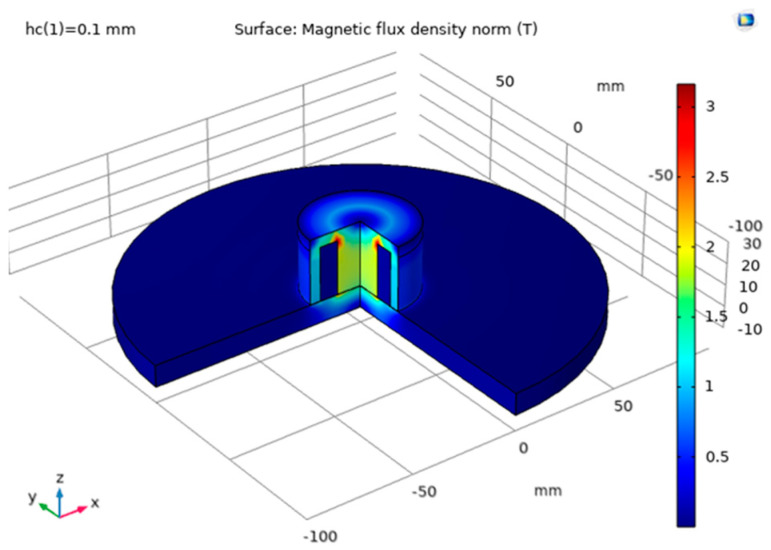
3D view of the magnetic flux density.

**Figure 4 sensors-22-09508-f004:**
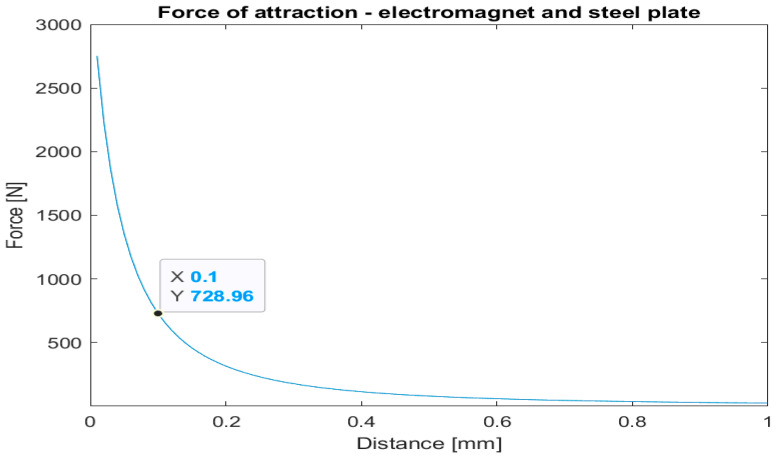
Force of attraction between steel plate and one electromagnet.

**Figure 5 sensors-22-09508-f005:**
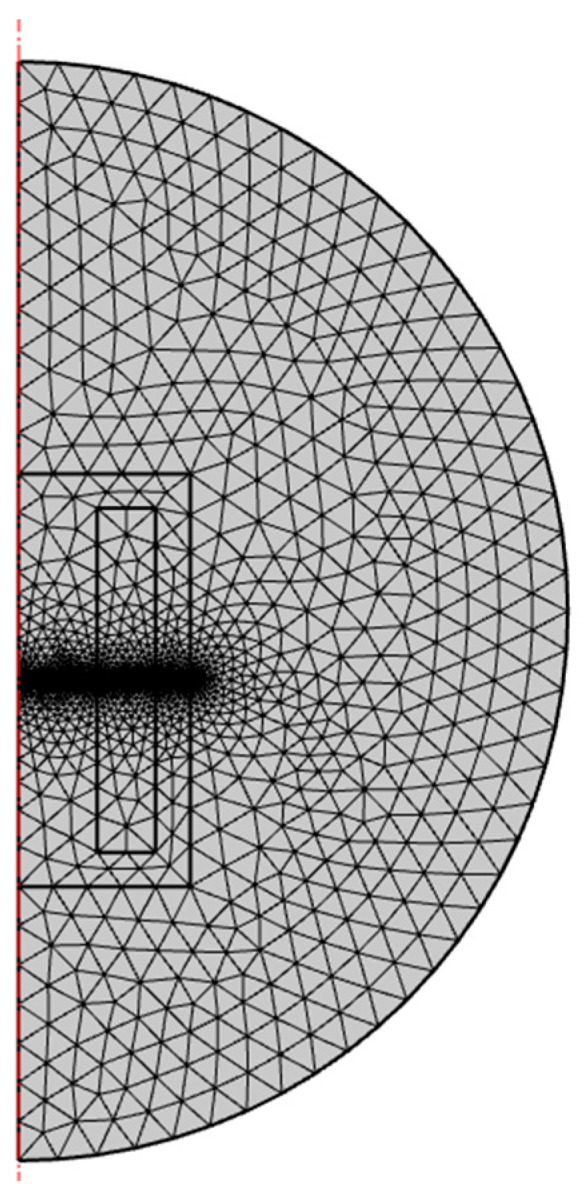
Cross-section of the measurement mesh during simulation.

**Figure 6 sensors-22-09508-f006:**
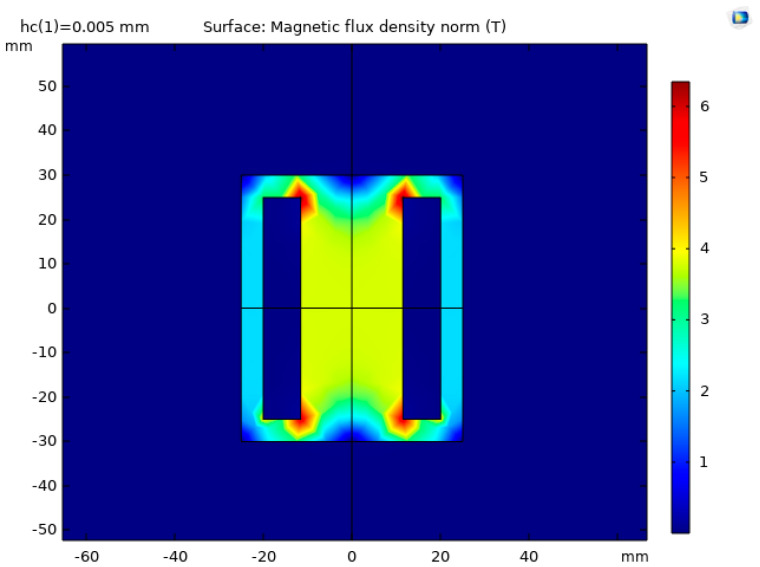
Cross-section view of magnetic flux density during simulation of two magnets.

**Figure 7 sensors-22-09508-f007:**
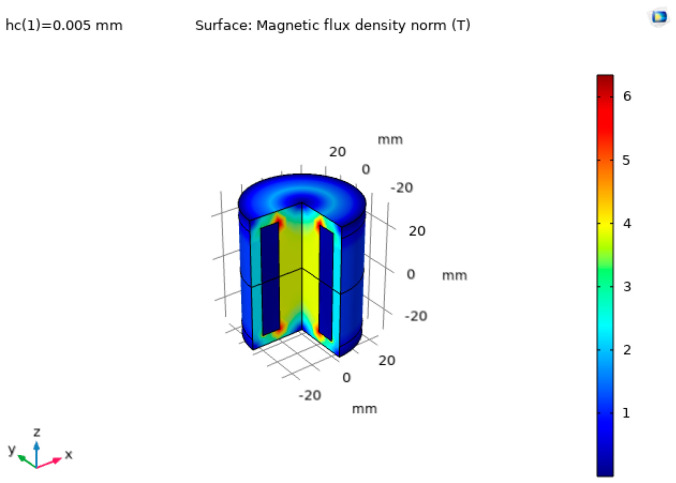
3D view of the magnetic flux density distribution in a simulation of two magnets.

**Figure 8 sensors-22-09508-f008:**
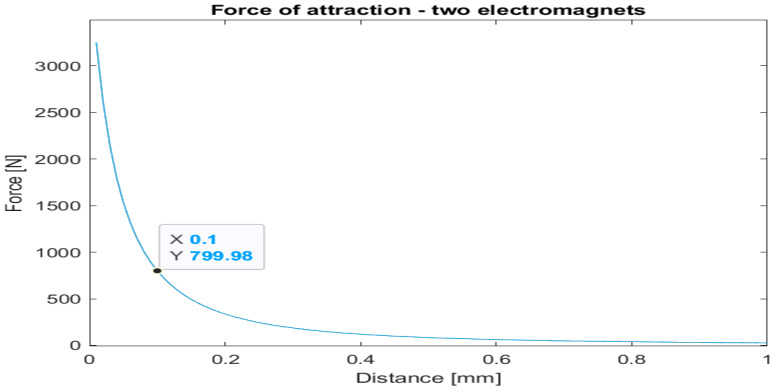
Results of two magnet simulation.

**Figure 9 sensors-22-09508-f009:**
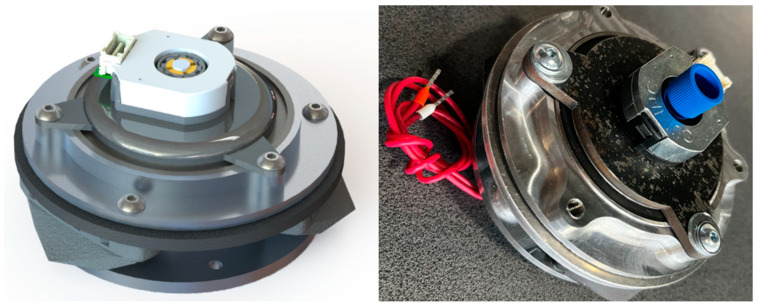
Integrated joint (CAD visualization and real-life device).

**Figure 10 sensors-22-09508-f010:**
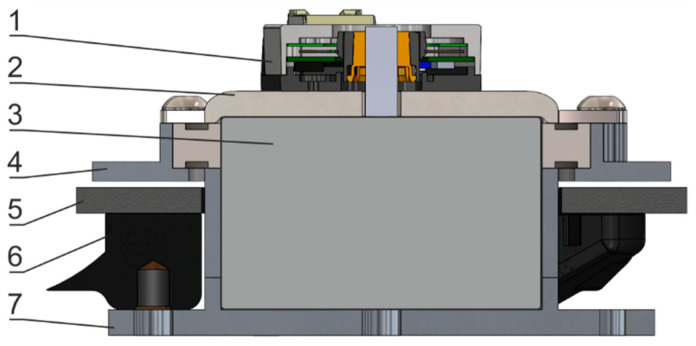
Cross-section of an integrated joint. 1—encoder, 2—steel plate, 3—electromagnet, 4—ball bearing housing, 5—brake disc, 6—brake pad, 7—electromagnet housing.

**Figure 11 sensors-22-09508-f011:**
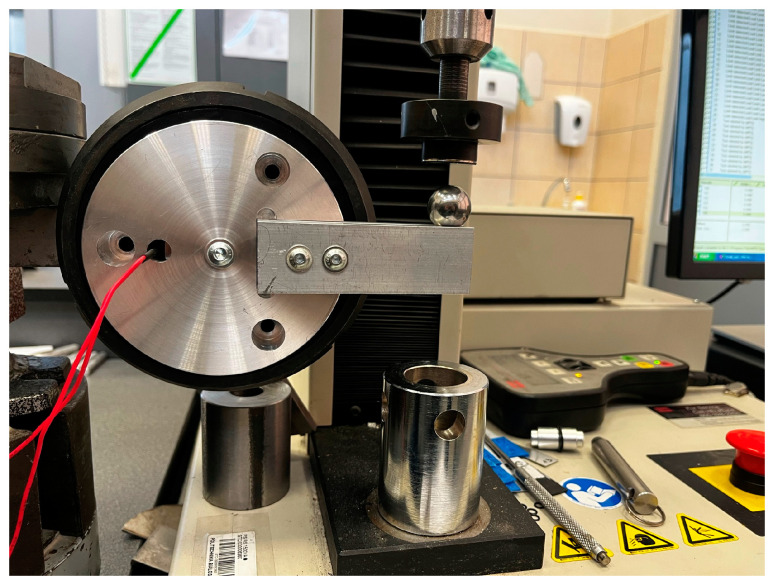
Test rig.

**Figure 12 sensors-22-09508-f012:**
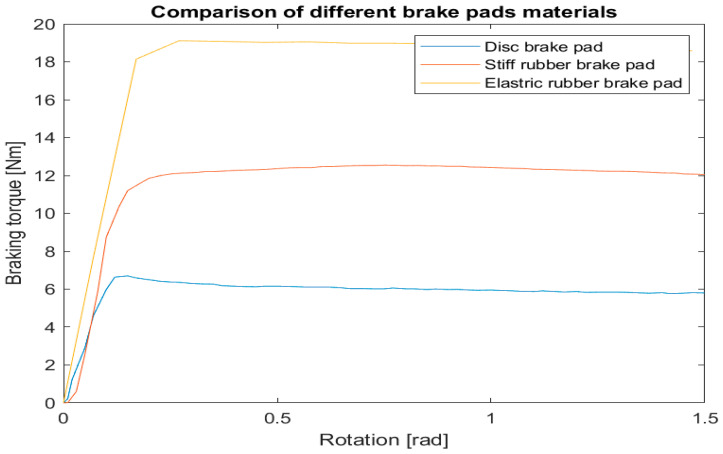
Comparison chart of braking torque of the tested brake pads materials—5 mm/s.

**Figure 13 sensors-22-09508-f013:**
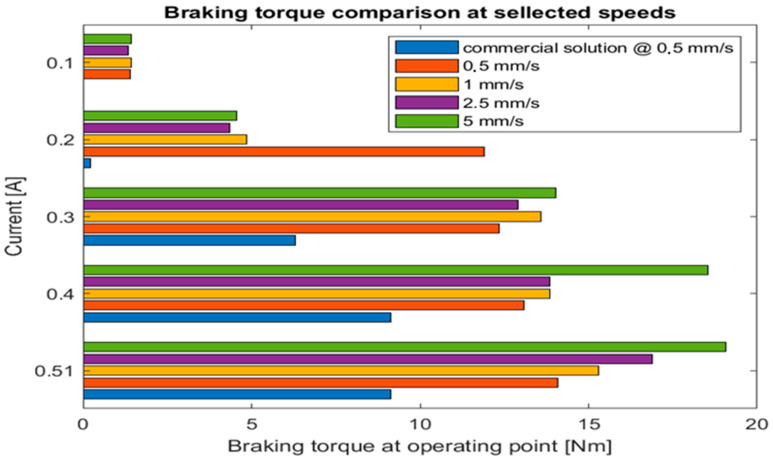
Braking solutions comparison (our design vs. commercial) of braking torque at different operating speeds in a point of operation regarding chosen current levels supplied to the electromagnet.

**Figure 14 sensors-22-09508-f014:**
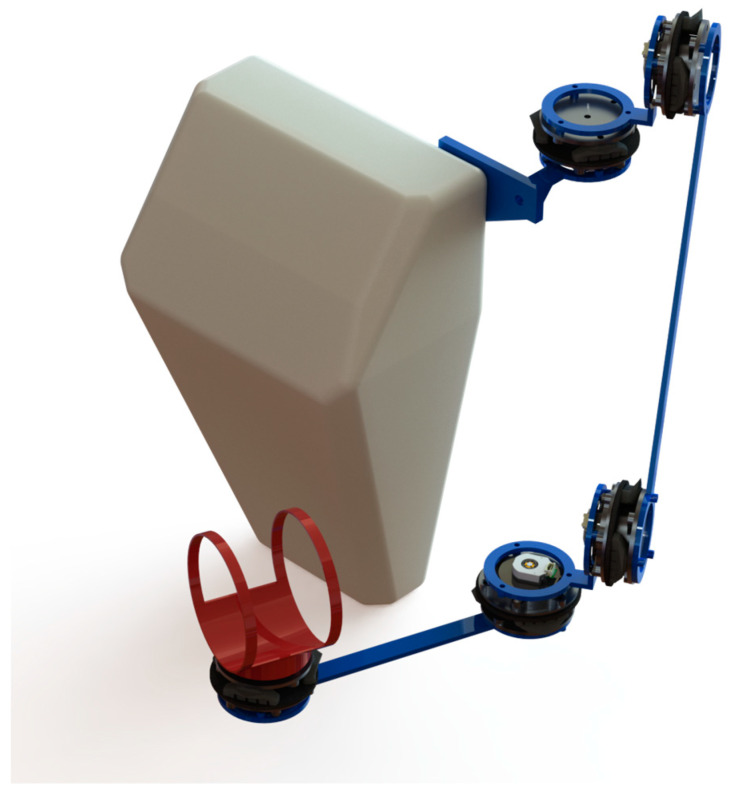
The proposed phantom device.

**Table 1 sensors-22-09508-t001:** Electromagnet characteristic.

Parameter	Value [Unit]
Manufacturer’s description/model	LS-P50/30
Diameter	φ 50 [mm]
Height	30 [mm]
Weight	319 [g]
Voltage	12 [V] DC
Power consumption	11 [W]
Coil resistance	30.372 [Ω]
Coil inductance	0.86905 [H]
Coil current [used during simulation]	0.6 [A]
The rated force of attraction	60 [kg]

**Table 2 sensors-22-09508-t002:** Simulation parameters.

Parameter	Value [Unit]
Parameter	750
Relative permeability of structural steel (steel plate)	400
Relative permeability of soft iron (casing and core of the electromagnet)	700
Number of coils	0.26 [mm]
Wire diameter	0.1 [mm]
Default distance between objects	750

**Table 3 sensors-22-09508-t003:** Mesh properties.

Mesh Property	Value [Unit]
Geometric entity level	Entire geometry
Element’s quality measure	Skewness
Mesh vertices	37,296
Element type	Triangle
Number of triangles	74,411
Edge elements	3853
Vertex elements	16
Minimum element quality	0.0796
Average element quality	0.8602
Element area ratio	4.59 × 10^−7^
Mesh area	10,050 [mm^2^]

**Table 4 sensors-22-09508-t004:** Mesh properties during simulation of two electromagnets.

Mesh Property	Value [Unit]
Geometric entity level	Entire geometry
Element’s quality measure	Skewness
Mesh vertices	3192
Element type	Triangle
Number of triangles	6282
Minimum element quality	0.5266
Average element quality	0.8435
Element area ratio	1.774 × 10^−4^
Mesh area	10,040 [mm2^2^]

## Data Availability

Not applicable.

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
