# Peer review of "A Cost-Effective, Integrated Haptic Device for an Exoskeletal System"

_sensors, 2022, doi:10.3390/s22239508_

Round 1

Reviewer 1 Report

The author introduced an integrated rotational joint with torque feedback based on electomagnetics. They designed a structure to house one or two magnets to provide braking torques in need and claimed its usage for exoskeletal systems. 

I think the manuscript in its current form is not ready for publication and my detailed comments are:

1. the presented module should be evaluated in an exoskeletal system if exoskeleton is the key application as mentioned in the title and introduction.

2. haptic devices based on electromagnetics are not new. The authors should clearly discuss why existing solutions based on electromagnetics do not work well and fairly compare with them to highlight their contribution. If the structure or the layout of the magnets is new, I am expecting more analysis on the design parameters. So far, a COMSOL simulation does not provide sufficient information to readers. 

3. Is the braking torque controllable? How is the torque controlled? Is it an open-loop control? if the torque is controllable, features such as the controllable range, precision or time delay should be reported and analyzed. 

4. The proposed device is an actuator more than a sensor. I am not sure if the topic best fits the theme of the journal ‘sensors’.

Other comments:

- line 82 ‘applying the force applied to…’ should delete one ‘apply’

- line 128-129, why the other device is not used as the reference design

- line 159, the design with two magnets are not shown in fig.4

Author Response

We appreciate very much the constructive comments and suggestions provided by the reviewers. We have carefully revised the paper addressing every point raised by each reviewer. The significant changes made in the paper are marked in yellow highlighting. If the reviewer thinks we need further modification and is willing to help us, please specify the modification as much as possible. Once again, we will revise it strictly with expert opinions. Thank you very much for the expert's help checking and pointing out the shortcomings. The following summarises the changes made in the revised version of this paper.

The responses to the reviews are in the attached PDF file.

Author Response

(The authors gave the same response as above.)

Round 2

Reviewer 1 Report

The revised manuscript has major improvements compared to the first submission. I would also thank the authors for commenting and clarifying a few problems. After reading the response letter, I still have one main concern about the contribution of this work. 

The authors claimed that haptic devices based on electromagnetics is a relatively unexplored area, but I cannot accept this is the first attempt to design haptic devices using  electromagnetics. Maybe those following works are also related :

[1]Andrew H. C. Gosline and Vincent Hayward, Eddy Current Brakes for Haptic Interfaces: Design, Identification, and Control, IEEE/ASME TRANSACTIONS ON MECHATRONICS, 2008

[2]Hashim Iqbal and Byung-Ju Yi, Design of a New Bilayer Multipole Electromagnetic Brake System for a Haptic Interface, applied science, 2019.

[3]Hashim Iqbal and Byung-Ju Yi, Design and Experimental Verification of a 3-DOF Spherical Electromag- netic Brake for Haptic Interface, International Journal of Control, Automation and Systems, 2020

We may also other related research in this line. There must be some differences of this work in comparison with the above works. Therefore, a thorough comparison along with some fair discussion is still needed to highlight the contribution of this work. 

I also suggest the authors to check the figures and diagrams in the above related works, e.g., fig.2 in [3]. A schematic diagram to illustrate the principle and the structural design would be helpful to present the designed device. 

Other detailed comments: 

- I am not sure if the exoskeletal system is designed for rehabilitation. If so, it only provides damping force and cannot used for passive training. This should be mentioned. 

Author Response

We appreciate very much the constructive comments and suggestions provided by the reviewers. We have carefully revised the paper addressing every point raised by each reviewer. The significant changes made in the paper are marked in green highlighting.

If the reviewer thinks the manuscript requires further modification and is willing to help us, please specify the adjustment needed. Once again, we will revise it strictly with expert’s opinions.

We deeply appreciate reviewer’s help and directing our attention towards provided literature sources. After analysis of those documents, we decided to modify our manuscript in an introduction and summary section to incorporate a better explanation of our work’s contribution.

Reviewer 2 Report

Dear Authors, 

please make one more step of efforts to get the paper in a better shape. 

Showing Fig. 1 and Fig. 5 does not give useful information. In fact both figures can be skipped. All other figures can be improved a lot: font sizes are so bad that the aspect ratio has been changed when the figure was inserted. Fig. 9 lacks of markers in the image. I see there an encoder.

One more little step is needed to polish the paper.